# Managing Spinal Muscular Atrophy: A Look at the Biology and Treatment Strategies

**DOI:** 10.3390/biology14080977

**Published:** 2025-08-01

**Authors:** Arianna Vezzoli, Daniele Bottai, Raffaella Adami

**Affiliations:** Department of Pharmacological Science, University of Milan, Via G. Balzaretti 9, 20133 Milan, Italy; ariannavezzoli@live.it (A.V.); raffaella.adami@unimi.it (R.A.)

**Keywords:** spinal muscle atrophy, spinraza, onasemnogene abeparvovec, risdiplam, survival motor neuron, curcumin, plastin 3, rehabilitation, psychological support

## Abstract

Spinal muscular atrophy is a very severe genetic disease that mostly affects newborns, toddlers, and young adults. It causes movement loss and, in some cases, death from respiratory failure. Currently, three pharmacological treatments are available to alter the disease course. However, improved and novel treatment strategies are necessary to improve patient outcomes and well-being.

## 1. The History of SMA Research

Research on spinal muscular atrophy (SMA) began in the late 19th century when two authors, Johann Hoffmann in 1893 and Guido Werdnig in 1891, separately described a genetic condition causing newborns to gradually lose muscle mass and become feeble. Classifications within the syndrome are based on different phenotypes, including the age at which it first manifests and the maximum motor function attained [1,2,3]. Two research teams linked a position on the long arm of chromosome 5 to chronic manifestations of the illness in the early 1990s [4,5]. However, functional ablation of the survival motor neuron (*SMN*) gene did not reproduce the spectrum of illness severity seen in human SMA. The centromeric gene *SMN2*, identified by researchers in the late 1990s, was determined to be essential in defining the severity of SMA, which is inversely correlated with the quantity of *SMN2* genes [6,7]. In the animal kingdom, the SMN protein plays a multifunctional role that is essential for survival. SMN affects a number of facets of RNA metabolism, controlling the synthesis of signal recognition particles, telomerase, small nuclear ribonucleoproteins (snRNPs), small nucleolar RNPs, and small Cajal body-associated RNPs. SMN expression also significantly impacts transcription, pre-mRNA splicing, histone mRNA processing, translation, selenoprotein synthesis, macromolecular trafficking, stress granule formation, cell signaling, cytoskeleton maintenance, and DNA repair [8]. Experiments on fruit flies, mice, and zebrafish have all contributed to our understanding of the disorder, improving genetic screening, allowing testing for possible treatments, and establishing how SMA affects the body.

These foundational experiments led to the development of medications for SMA patients. The US Food and Drug Administration (FDA) and the European Medicines Agency (EMA) currently approve three treatments: nusinersen, the first medication to treat SMA (Spinraza, Biogen; FDA approval in 2016 and EMA approval in 2017); onasemnogene abeparvovec (Zolgensma, Novartis; FDA approval in 2019 and EMA approval in 2020); and risdiplam (Evrysdi, Roche; FDA approval in 2020 and EMA approval in 2021).

Research into many alternate approaches for SMA treatment is ongoing, targeting muscle, metabolic activity, and oxidative stress. The evidence base for these new approaches is relatively robust, especially for young patients; they will be discussed in a later section as future research directions.

## 2. Key Genetic Evidence for SMA

Early diagnosis, effective treatment planning, and public health initiatives all depend on an understanding of the prevalence, incidence, distribution, and risk factors of SMA. SMA impacts people of all ethnic backgrounds [9]. The autosomal recessive pattern of inheritance means a child must inherit two faulty copies of the *SMN1* gene, one from each parent, to develop the condition [10]. Carrier screening programs have been implemented to allow the provision of genetic counseling, particularly in regions where SMA is more prevalent. Some research suggests that SMA frequency and prevalence can vary by geographic area and are correlated with ethnicity [11]. Since their introduction in various nations, newborn SMA screening programs have significantly altered the disease’s epidemiology. Prompt intervention made possible by early diagnosis can improve the quality of life and outcomes for those impacted [12,13,14,15]. The estimated incidence of SMA is around 0.8 in 10,000 live births [9,16,17], while the prevalence is 1–2 in 100,000 persons [11]. Indeed, it is the most common rare disease affecting the young [18]. While there are notable regional differences in frequency, these are most likely related to small sample sizes and patient populations. Regions with robust health systems and greater knowledge of neuromuscular illnesses may report higher incidence rates, as is the case in Sweden. Incidence, which measures the number of new instances of the disease, shows relatively high variance; on average, it is around 8 in 100,000 live births. There is a great deal of variation in this parameter between nations, with high levels particularly common in islands with small populations like Iceland. Such variance may also be the result of inadequate data analysis, as was the case in a study conducted in Slovakia that discovered an incidence of 17.8 per 100,000 people [11]. It may also be true, though, that different ethnic groups have varying levels of access to healthcare. A high degree of consanguinity may help to explain the high incidence (24 per 100,000 live births) seen in a small research study of 75,000 people in Libya [19].

Numerous genes—including those for *SMN*, neuronal apoptosis inhibitory protein (NAIP), and p44, which codes for a transcription-factor component—are found in the critical chromosomal area of 5q13 [20]. This region is unstable and prone to rearrangement, as demonstrated by the numerous band patterns seen in pulsed-field gel electrophoresis investigations employing probes for *SMN1* and the neighboring *NAIP* [21] telomeric region. Although the main determinant of SMA severity is *SMN*, other genes may nevertheless play minor roles; they are also reported to be mutated in many SMA cases, particularly *NAIP* [22]. Human SMN is a protein composed of 294 amino acids found in both the cytoplasm and the nucleus [23,24]. The SMN protein is part of a bigger complex called Gemins, which is made up of a number of other proteins (Gemin2–8) and UNRIP [25]. Additionally, it has been shown that SMN and the SMN complex may interact momentarily with a wide range of other proteins and RNA, such as hnRNPQ/R [26] and cytoskeleton proteins like actin and profilin [8,27,28]. The biogenesis, assembly, and nuclear trafficking of snRNPs are all significantly impacted by this complex [29,30].

Humans have two copies of the centromeric *SMN1* gene and a variable number of telomeric *SMN2* genes [6,31,32]. Every animal species that has been studied possesses the *SMN1* gene [33]. Chimpanzees contain two to seven copies of the *SMN* gene per diploid genome [34], while other species, such as mice, only have two [35]. In humans, there are just 11 nucleotide differences between the *SMN1* and *SMN2* genes. Remarkably, a C6T transition in exon 7 functions as an exonic enhancer but does not affect the amino acid sequence; instead, it produces a shorter form lacking exon 7 [36] in around 90% of instances. About 10% of the *SMN2* gene is translated into a full-length protein that can partially compensate for deficiencies in *SMN1* [2].

Subtle alterations of the second allele of chromosome 5, alongside *SMN1* deletion, are detected in around 4% of SMA cases [35,37]. Two minor *SMN1* variations have been identified, although they are extremely uncommon and exclusively occur in consanguineous families (23). At least 108 distinct pathogenic mutations of *SMN1* have been identified [38,39,40]. The 5q13 area is very unstable and vulnerable to uneven gene conversion and recombination. This frequently results in de novo mutations, which are less commonly caused by gene conversion events and more often by uneven recombination, accounting for 2% of SMA cases [41,42]. This extensive genetic knowledge has contributed to the development of reliable procedures to test for SMA.

Since the early start of treatment greatly improves long-term prospects, prompt and precise diagnosis of SMA is essential. Clinical assessment is usually the first step in diagnosing SMA after signs like hypotonia, muscular weakness, or delayed motor milestones are noticed, especially in newborns and young children [17]. However, genetic testing is necessary to confirm the diagnosis, as these symptoms can mimic those of other neuromuscular conditions, including congenital myopathies and amyotrophic lateral sclerosis. Testing for the homozygous deletion of exon 7 in the *SMN1* gene is the most precise diagnostic technique for identifying SMA [43]. Digital PCR (dPCR), multiplex ligation-dependent probe amplification (MLPA), and quantitative polymerase chain reaction (qPCR) are common techniques. For the diagnosis of SMA, the homozygous deletion of *SMN1* is specific, while the copy number of *SMN2* is essential for clinical classification, disease severity, and prognosis. Carrier screening is a fundamental component of SMA care, particularly for providing genetic counseling to at-risk families. Prenatal testing performed in couples who are carriers is another essential technique for diagnosing SMA. Amniocentesis and chorionic villus sampling are two methods utilized to identify SMA in fetuses [44,45].

## 3. Key Phenotypic Evidence for SMA

Regardless of the degree of phenotype, the number of *SMN2* copies is used as a predictive tool to guide treatment approaches and care plans for people with SMA. It also represents an ideal target for treatment development.

Given their high metabolic needs and the lengthy axonal transport necessary for optimal function, motor neurons (MNs) appear to be the cells most affected by mutations in the *SMN1* gene, and are perhaps the best understood. SMN may be protective for MN growth cone activity, intracellular transport, endocytosis, and autophagy [18,46]. However, it is not entirely clear how SMN protein deficiency results in MN mortality. MNs are a unique cell type because of their size, often with very long processes that necessitate efficient transport apparatus [47,48]. This apparatus likely requires functioning SMN more than other cells, due to its diverse roles [18]. However, SMA also affects other organs. For instance, it can cause heart anomalies—usually atrial and ventricular septal defects—which frequently occur at the same time as changes to the autonomic nervous system and the normal heart rhythm, and increase the risk of sudden death [35]. SMA also has a significant impact on muscles themselves; as a matter of fact, in animal models with selective deletion of *SMN1* in muscle tissue, fiber necrosis is observed [49]. The SMN protein interacts with others, such as α-actinin and αB-crystallin, highlighting its vital role in the structure and functionality of muscles. However, different studies have demonstrated that restoring SMN levels in neurons alone is sufficient to markedly increase survival and motor function in SMA models, despite the extremely serious pathogenic effects of SMN loss in muscle [47].

Patients with all types of SMA suffer from respiratory insufficiency, which increases along with the disease severity; deep tendon reflexes are either missing or severely diminished, and muscular weakness is present. The characteristic bell-shaped chest and paradoxical breathing pattern present in the most severe forms are caused by weakness of the intercostal muscles [50].

SMA patients can be divided into eight different phenotypes (1A, 1B, 1C, 2A, 2B, 3A, 3B, and 4) [47,51]. However, these traditional SMA types by themselves are insufficient for identifying patient populations. The most significant predictive variables for a symptomatic patient’s response to treatment are age at onset, length of disease, and motor function level at the beginning of treatment [52]. Of SMA 1 patients, 96% had homozygous deletions in *SMN1*, 94% in SMA 2, and 86% in SMA 3 [40,53].

The most severe and potentially fatal form of SMA, known as SMA 0 (or SMA 1A), manifests prenatally as severe weakness, hypotonia, and decreased fetal movement in late pregnancy. Neonates with this condition have cardiac abnormalities, areflexia, facial diplegia, and occasionally arthrogryposis. Respiratory failure usually results in death within the first 6 months [54,55]. The most common variety of spinal muscular atrophy is SMA 1 (Werdnig–Hoffmann disease), which accounts for around 50% of all instances of the ailment that are identified [56]. Alongside type 1A, these patients can also be categorized as 1B or 1C. SMA 1B patients are usually born with insufficient or nonexistent head control and display clinical symptoms before the age of 3 months. Patients with SMA 1C have some head control and survive longer than those with the 1B form, who usually die by the age of two [51,54]. Constipation, delayed stomach emptying, dysphagia, vomiting, and gastroesophageal reflux are common gastrointestinal symptoms in these patients. These variables all have a significant role in increasing the risk of aspiration and pneumonia, which continues to be the leading cause of death in this group. Malnutrition results from a combination of masticatory muscle weakness, dysphagia, and emerging respiratory issues that reduce caloric intake [57]. It has also been reported that milder forms of SMA do not produce axonal degeneration of sensory neurons, whereas severe forms do [58].

Individuals with SMA 1 have cardiac anomalies (Figure 1) that are not seen in individuals with milder forms [59]. These include distal digital necrosis of the blood vessels and bradycardia septal defects (Figure 1) [60,61]. Additionally, newborns with SMA 1 typically show hyperplasia of the pancreatic islet α-cells, linked to aberrant glucose levels [62] (Figure 1 and Table 1). High levels of serum leptin and disruptions in the metabolism of fatty acids are examples of further metabolic dysfunction [63].

In SMA 2 and 3, gene conversion of *SMN1* results in an elevated *SMN2* copy number (1–8 copies of the *SMN2* gene are observed in the healthy human genome) [64], whereas most SMA 2 patients have a full *SMN1* deletion. Incomplete gene conversion produces a hybrid *SMN1/SMN2* gene, with exon 7 of *SMN2* origin and exon 8 of *SMN1* origin.

*SMN2* copy number is the most significant predictor of clinical severity and age of onset in presymptomatic SMA 2 patients [52]. Usually diagnosed at 6 months of age, infants with SMA 2 are able to sit up on their own, make it through puberty, and typically go on to adulthood. SMA 2A patients may lose the ability to sit, and have increased scoliosis and intercostal muscular weakness that manifests in restrictive lung disease (Figure 1) [65]. SMA 2 accounts for 20% of all cases [56] (Table 1).

Known as Kugelberg–Welander disease, SMA 3 is seen in about 30% of SMA patients. There are two subtypes: SMA 3A manifests between the ages of 18 months and 3 years, while SMA 3B manifests between the ages of 3 and 30 years [56]. At first, patients may walk without assistance, but they typically acquire very severe scoliosis. Although they often live normal lives, they lose the ability to walk at a young age. In comparison to those with SMA 3A, these patients are able to walk unassisted for a longer time [35]. Obesity and other mobility-related issues may emerge in these patients [66] (Table 1).

Usually, SMA 4 develops in adulthood. The illness course is very mild—frequently undetectable—and patients often have a normal life expectancy [56,65] (Table 1).

Given that the majority of developed countries have access to causative therapy, the previous definitions of SMA types are a little outdated. A new classification, focusing on newborn screening and treatment options, was recently published [67]. This new approach is derived from newborn screening itself. The authors divided patients into three classes:Clinically undetermined and genetically determined;Clinically and genetically determined infants;Clinically determined but genetically undetermined infants.

They suggest appropriate pharmacological treatments based on classification, including nusinersen, onasemnogene abeparvovec, and risdiplam.

## 4. Therapeutic Approaches

### 4.1. Biotechnological Treatments

It is interesting to note that approved treatments and those currently being studied are all based on innovative approaches, such as RNA technologies, gene therapies, exon modifiers, and antibodies.

### 4.2. Antisense Oligonucleotides

A family of genetically based medicines known as antisense oligonucleotides (ASOs) holds great promise for treating progressive genetic illnesses for which there is currently no cure. This includes genetic neurodegenerative diseases, such as Huntington’s disease, SMA, and amyotrophic lateral sclerosis [68]. By targeting almost any RNA sequence, ASOs can induce a variety of effects, including gene knockdown and splicing control of pre-mRNAs through a steric block mechanism [69].

The first effective medication developed for SMA was nusinersen, which Biogen markets as Spinraza. It was authorized by the U.S. FDA in December 2016 and the EMA in 2017 [70] (Table 2).

Nusinersen is an ASO that promotes exon 7 inclusion by binding to the intronic splicing inhibitor in intron 7 of *SMN2* mRNA transcripts [71] (Figure 2). Nusinersen selectively attaches itself to one intronic splice silencing site (ISS-N1) in *SMN2* pre-mRNA’s intron 7. Splicing components that would normally bind and prevent splicing are dislocated by this binding activity. Exon 7 is therefore incorporated into the *SMN2* mRNA and translated into a full-length, functional SMN protein. Successful treatment has been shown both in a transgenic mouse model of SMA and in human clinical trials, as evidenced by significant increases in full-length *SMN2* pre-mRNA and greater levels of SMN protein in MNs [72,73].

Nusinersen is available as an injectable solution and is authorized for use in both adult and pediatric patients.

MNs in the spinal cord and brain stem are the first cells affected in SMA. In order to increase the likelihood that the treatment will specifically target the most impacted cells, nusinersen must be injected into the spinal fluid by means of a lumbar puncture while the patient remains still; it cannot pass the blood-brain barrier due to its high molecular weight. Furthermore, children may need adjunct medications, and interventional radiology is required for patients with severe scoliosis or a history of previous spinal fusion to provide imaging guidance for treatment [74].

Nusinersen is metabolized via hydrolysis and catalyzed by exonucleases. In the cerebrospinal fluid, the typical terminal elimination half-life is predicted to be between 135 and 177 days; for this reason, the administration interval is 120 days. Urinary excretion of nusinersen and its metabolites is the main method of elimination [75].

The ENDEAR study, a double-blind controlled clinical trial, demonstrated the efficacy of Spinraza. Open-label clinical trials in both presymptomatic and symptomatic SMA patients (the Cherish, Nurture, Shine, and Embrace studies) have substantiated this efficacy and offer further information on the treatment’s safety [76,77].

Compared to a placebo group, 51% of babies with infantile onset 5q SMA treated with nusinersen exhibited improved neuromuscular functioning, attained exceptional motor milestones, and had a higher probability of survival. In stark contrast to the untreated group’s decline in motor abilities, even individuals with later-onset 5q SMA variants exhibited improvements in their motor functions. Indeed, nusinersen is beneficial for a wide range of 5q SMA phenotypes, as evidenced by the patients’ maintenance of motor milestones over time. The most noticeable therapeutic advantages were observed when nusinersen was administered early in the course of the illness (NURTURE trial) [72].

A different study evaluated the safety and effectiveness of nusinersen in adult patients with genetically verified 5q-associated SMA over the course of 38 months. This data offers further empirical support for nusinersen’s long-term effectiveness and safety [78].

### 4.3. Gene Therapy

Recombinant DNA technology, which involves introducing a gene of interest into a vector (such as a plasmid, nanostructured construct, or viral vector), is one of the most common genetic treatment techniques. Gene therapy has many inherent obstacles; one of the most significant is the challenge of getting the gene into the cell. Gene transfer takes advantage of a molecular carrier. After being administered to the patient, the vector should not trigger any inflammatory or allergic reactions [79] (Figure 3). Since a viral vector can efficiently pass through the cell membrane and transfer a genetic component, this is a frequently utilized delivery technique. The two steps in the gene therapy process are integrating a functioning gene into the genome and replacing the aberrant, disease-causing gene.

The gene either lives as an episome or integrates with the patient’s genome after entering the cells. When the cellular machinery recognizes the transferred gene, it starts the process of producing the encoded protein. Correcting dysfunction brought on by the mis-encoded gene requires this product.

In 2019 and 2020, respectively, the FDA and EMA authorized onasemnogene abeparvovec, sold under the name Zolgensma, a gene replacement therapy for *SMN1*. An AAV9 viral vector is used to introduce a functional copy of the human *SMN1* gene into MN cells [80] (Table 2).

The pharmaceutical formulation—a suspension for intravenous administration—can only be administered at a medical facility under the guidance of a licensed physician.

According to Ogbonmide and colleagues [80], the suggested dose for onasemnogene abeparvovec is 1.1 × 10^14^ vector genomes (vg) per kilogram of body mass, given as a single infusion for 1 hour. Clinical trials have demonstrated the effectiveness of onasemnogene abeparvovec in treating babies with SMA between the ages of 2 weeks and 8 months and weighing less than 13.5 kg [80] In a recent analysis, onasemnogene abeparvovec was found to be effective in a UK population between 3.2 and 20.2 kg and aged between 0.6 and 89 months [81].

### 4.4. Splicing Modifier

The FDA authorized risdiplam (Evrysdi), a modifier of *SMN2* splicing [82], as the first oral medication to treat SMA in 2020, and the EMA approved it in March 2021 for children older than 2 months [83] (Figure 4 and Table 2). Risdiplam transforms the weak 5′ splice site of *SMN2* exon 7 into a stronger variant by stabilizing the transitory double-stranded RNA structure created by the U1 snRNP complex and the 5′ss of *SMN2* exon 7 [84]. After oral administration, risdiplam is quickly absorbed, and peak plasma concentrations are observed 1 to 4 h after.

The efficacy of risdiplam has been rigorously evaluated in several clinical trials, including the FIREFISH and SUNFISH studies. Infants with SMA 1, who typically have significant motor disability and a high death rate, were the primary focus of the FIREFISH Trial. Significant clinical improvements were demonstrated by the achievement of important motor milestones, such as sitting alone, which are rarely observed in untreated patients. Treatment also increased their survival rate without the need for ongoing mechanical ventilation [85]. The open-label extension of FIREFISH research from the risdiplam trial demonstrated that treatment remained safe and effective in patients with SMA 1 over a 5-year period, as presented during the 2024 Cure SMA Research and Clinical Care Meeting. By the end of year 5, 91% of the children who received the therapy were still alive, greatly exceeding previous survival rates.

Risdiplam has proven to significantly enhance SMA 1 patient outcomes. According to a Matching-Adjusted Indirect Comparison (MAIC) approach, risdiplam has a lower incidence of serious adverse events (AEs), better survival rates, and improved motor function than nusinersen. Despite strong results in favor of risdiplam, the variety of research populations has made it difficult to draw firm conclusions when comparing the treatment with onasemnogene abeparvovec using Synthetic Treatment Comparison (STC) methods. The difficulty of cross-comparing therapies across various SMA subtypes was further highlighted by the lack of definitive results from MAIC analysis of risdiplam vs. nusinersen in SMA 2 and 3 patients [86].

According to the Motor Function Measure (MFM32) grading scale, risdiplam significantly improves motor function in older patients with SMA 2 and 3, as demonstrated by the SUNFISH trial. The results were consistent over an extended length of time and for a broad age range from childhood to adulthood [87]. According to Roche guidelines, risdiplam can be administered at an age lower than 2 months (0.15 mg/kg/day) in a liquid formulation, for up to more than 2 years (more than 20 kg of weight and a dosage of 5 mg/day in liquid or tablet formulation); this dosage can also be administered to adult patients (https://www.evrysdi-hcp.com/content/dam/gene/evrysdi-hcp/pdf/evrysdi-hcp-dosing-guide.pdf, accessed on 12 May 2025).

### 4.5. Ongoing Trials of Approved and New Molecules

Although the three aforementioned approaches have strongly contributed to the health of SMA patients—especially those with SMA 1—basic and clinical research has continued.

For instance, the FDA authorized a new formulation of risdiplam for administration in tablet form. Additionally, a case study pushed for the use of risdiplam starting in the 32nd week of pregnancy [88]. Just 2 months later, a new paper on the intra-amniotic delivery of nusinersen in two SMA mouse models showed that when compared to animals treated postnatally (between P1 and P3), prenatal therapy enhanced MN counts, motor axon development, motor behavioral tests, and survival in SMAΔ7 mice, who exhibit pathology in utero [89].

Trials that are close to FDA approval and are currently in phase 3 are for high doses of nusinersen and intratracheal treatment with AVXS 101. The latter is directed towards young patients, aged between 2 and 6 years or 6 and 18 years, who had undergone previous treatments with at least four loading doses of nusinersen or at least 3 months of treatment with risdiplam. This trial expands the results of previous work on onasemnogene abeparvovec, which only included SMA 1 patients.

More recently, the FDA granted a priority review for a Biologics License Application (BLA) for apitegromab as a treatment for SMA. The FDA concluded that apitegromab might significantly improve the safety or efficacy of SMA treatment if authorized, as indicated by their priority review designation [90,91].

Apitegromab (Study SRK-015-003) is an experimental, completely human immunoglobulin G4 monoclonal antibody that prevents myostatin from activating. Apietegromab blocks the release of the mature myostatin—a transforming growth factor β superfamily–signaling protein that adversely affects skeletal muscle development and strength—by binding preferentially to both pro- and latent forms [92]. This is associated, in general, with improved muscle mass and strength.

RO7204239 is a monoclonal anti-myostatin that was developed and studied by Hoffmann-La Roche, for which a phase 3 trial is ongoing. It is a potent antibody that targets human latent myostatin, inhibiting its conversion to the active form when administered subcutaneously. This muscle-directed treatment is administered in combination with risdiplam (RO7034067) to investigate the safety and efficacy in SMA patients with walking capabilities (MANATEE study, NCT05115110).

The latest pharmacological treatments for SMA gradually target a variety of cellular and molecular targets because of the disease’s complexity. Various theories have been proposed and are now being researched in an attempt to create more effective SMA treatments, as SMN’s diverse functions in cells demand a multitarget approach.

### 4.6. Alternative Approaches

In recent years, natural substances and small medicinal molecules have also been used to treat SMA. An important example is curcumin, a polyphenolic substance found in turmeric that has demonstrated potential in raising SMN protein levels by concurrently reducing oxidative stress and inflammation. The substance also induces apoptosis by blocking important signaling pathways like mitogen-activated protein kinase (MAPK) and nuclear factor kappa-light-chain-enhancer of activated B cells (NF-κB) [16]. MN degradation in SMA is known to be exacerbated by oxidative stress and inflammation. Oxidative stress is a harmful process observed with the build-up of reactive oxygen species (ROS). ROS are especially harmful to cells in areas that require a rich oxygen supply, such as those in the nervous system. Oxidative stress affects the splicing machinery of the *SMN1* and *SMN2* genes [93]; furthermore, certain promoter sequences play a significant role in regulating the splicing of *SMN* exon 7 in oxidative stress conditions [94,95]. Seo and colleagues induced oxidative stress in an animal model of SMA, namely a transgenic (TG) mice with an FVB/N background, which have two copies of *SMN2* and one copy of Smn (Smn^+/−^; SMN2^+/+^) [95]. Their results suggest that several cis-elements and transacting factors are implicated in the splicing control of distinct *SMN* exons under normal and oxidative stress conditions, as well as in the susceptibility of different *SMN2* exons, including the 7, to skipping under OS conditions. It has been demonstrated that curcumin causes the nuclear translocation of nuclear factor erythroid 2 (NRF2), the primary transcription factor regulating the antioxidant responsive element. Curcumin improves the induction of neural stem cells (NSCs) into MNs and their resistance to oxidative stress, offering encouraging prospects for neuroprotective therapies. The relationship between curcumin and NRF2 is particularly noteworthy, since it affects how the SMN protein is expressed. NRF2 increases the synthesis of SMN, most likely via binding to the antioxidant responsive element in the promoter region of the *SMN1* gene [96]. Additionally, curcumin positively affects other critical characteristics of NSCs, including their capacity to multiply and self-renew in order to restore destroyed MNs [96]. Curcumin’s limited bioavailability seriously hinders clinical utilization, despite its promising therapeutic properties. Because of its very low solubility and quick metabolism, oral dosing shows less than 1% of this drug entering systemic circulation [16]. A number of approaches have been proposed to address this issue, including co-administration with the glucuronidation inhibitor piperine, the use of chemically modified derivatives, the use of delivery systems based on nanotechnology, and non-oral delivery. Curcumin is well tolerated in humans at a dosage of 1–4 g/day, according to preclinical research, including in the context of Alzheimer’s disease [16].

A number of other studies conducted in recent years have focused on muscle loss treatments aimed at blocking myostatin (growth differentiation factor 8; GDF-8), a protein that negatively regulates muscle growth. As a member of the transforming growth factor-β superfamily, myostatin may contribute to muscle atrophy and wasting and could be an accessible therapeutic target for conditions like SMA. Biohaven Pharmaceuticals is now researching the effectiveness of taldefgrobep-α, a new myostatin inhibitor, in enhancing muscle mass and strength in SMA patients [97].

The two main mechanisms of action of taldefgrobep-α are the blocking of signaling pathways linked to the activin receptor type 2B (ActRIIB) and the inhibition of myostatin function. This dual action distinguishes the medication from other myostatin inhibitors currently on the market. A phase 3, worldwide, randomized, double-blind, placebo-controlled study called RESILIENT (NCT05337553, still ongoing) is being conducted to assess the safety and effectiveness of the experimental medication as a supplement to common SMN upregulators like risdiplam or nusinersen [83,98,99].

In mild SMA models, notable gains in muscle mass and functional ability were demonstrated after recent developments in gene therapy employing AAV vectors to carry soluble ActRIIB or myostatin pro-peptides resistant to proteolytic activation [100,101].

Neuromuscular junction (NMJ) dysfunction, defined by anomalies in NMJ formation, maturation, and overall function, is significantly linked to the muscle weakening and increased fatigue associated with SMA. SMA disrupts the agrin/MuSK signaling pathway, which is essential for the development and maturation of NMJs. The enhancement of NMJ stability and functionality has been the focus of particular agrin-based treatment therapies. In SMA mouse models, the overexpression of agrin [102] enhanced NMJ architecture and reduced the severity of the disease. Additionally, it was demonstrated that subcutaneous administration of the active agrin fragment NT-1654 to SMA mice slowed the progression of the disease. The utilization of several therapeutic approaches aimed at improving cholinergic transmission is still being considered for SMA 2 and 3, as NMJ dysfunction has a significant effect on fatigue along the course of the disease [83].

A well-known medication that inhibits acetylcholinesterase, pyridostigmine, is primarily used to treat myasthenia gravis. Its use increases cholinergic transmission by slowing the breakdown of acetylcholine in the synaptic cleft. Preliminary findings from a phase II clinical trial investigating the effects of pyridostigmine in SMA patients showed decreased fatigability, supporting the medication’s use as an adjuvant treatment [103].

In SMA, NMJ nerve terminals exhibit decreased calcium signaling, which further inhibits the release of neurotransmitters. In SMA animal models, new treatments like the Ca(v)2.1/Ca(v)2.2 channel modifier, R-roscovitine, and a cyclin-dependent kinase 5 (Cdk-5) inhibitor that prolongs voltage-gated calcium channel openings, have been demonstrated to enhance NMJ construction and boost survival [104].

Loss of SMN triggers the p53 cell death pathway, which is thought to be crucial for MN degeneration in severe SMA models. The subsequent reduction of Stasimon, a transmembrane protein found in the endoplasmic reticulum, due to SMN-mediated U12 (an snRNA gene) intron splicing may be linked to the activation of p53. However, Stasimon overexpression did not reverse the SMA phenotype [35].

Autophagy and endocytosis are important cellular mechanisms linked to the pathogenesis of SMA. Such modulating proteins include Plastin 3 (PLS3), an actin-bundling protein; Neurocalcin delta (NCALD), a neuronal calcium sensor and a suppressor of the endocytic process; Coronin 1C, a conserved actin-binding protein; and Calcineurin-like EF-hand protein 1 (CHP1).

As evidenced by the reduction of MN pathology and enhanced NMJ structure and function in different SMA models overexpressing PLS3, an emphasis on endocytosis may result in promising combination therapies specifically targeting milder forms of SMA; in severe cases, however, the effectiveness of such approaches has remained limited [105]. Similar therapeutic advantages have been attained by lowering NCALD protein levels; these results have been replicated with comparable therapeutic benefits [106]. The ubiquitin-proteasome system (UBA), a crucial mechanism for protein degradation, is also impaired in SMA. In preclinical research, functional restoration of the *UBA1* gene has demonstrated encouraging outcomes, establishing it as a potent SMN-independent therapeutic target. The ubiquitin-activating enzyme E1, encoded by *UBA1*, is an essential part of the ubiquitin-proteasome system (UBA), which breaks down undesirable or damaged cellular proteins [83].

The synthetic peptide MR-409, an agonist of the growth hormone-releasing hormone (GHRH) receptor, is a novel therapeutic agent that has demonstrated effectiveness in postponing the progression of SMA, improving motor function by reducing muscular atrophy, enhancing NMJ maturation, and reducing spinal cord inflammation in models. MR-409 is a promising candidate, particularly for use in conjunction with other SMA-treating drugs. However, MR-409 is not yet authorized for use in humans, and more research is required to get this substance to market [107].

Off-label, salbutamol, also known as albuterol in the US, is a β2-adrenoreceptor agonist that helps SMA sufferers increase their strength and resilience. Although more research is required to fully understand the mechanisms, preliminary findings indicate that it may play a role in NMJ function [108].

## 5. SMA Metabolism and Nutrients: Key Evidence

Baranello et al. conducted a cross-sectional study to look into how the body composition and motor function of children with SMA 1 and 2 compare. Better motor skills were linked to a healthy body composition in the 88-child study [109]. Specifically, the study showed that individuals with SMA had higher fat mass levels than healthy controls. A link was observed between lower fat-free mass and lower motor skills, indicating that body composition could be a useful indicator to determine the severity of SMA. A number of metabolic changes were also seen in these SMA children, which could make clinical care more challenging. Lipid profile dysregulation was the most notable change found [109]. Deguise et al. recommended cautious screening procedures and close monitoring of lipid profiles in SMA patients, speculating that denervation may be a contributing factor to the reported lipid abnormalities. Such a strategy might help provide customized nutritional treatments to address metabolic issues [110]. Several metabolic disorders, such as dicarboxylic aciduria, high excretion of urinary acylcarnitine, carnitine deficiency in both muscle and serum, and deficiency of acyl-CoA dehydrogenase—an enzyme crucial to β-oxidation processes—can result from fatty acid metabolism, which can be severely impaired in patients with SMA 1 [57]. About 37% of SMA patients are at higher risk of developing dyslipidemia, according to a recent study on the metabolic health of SMA patients. Pathological specimens from these individuals frequently exhibit liver steatosis. These results link the liver’s intrinsic SMN deficit to aberrant fat buildup within hepatocytes and the disruption of normal hepatocyte function. Patients with SMA may be more susceptible to more severe types of liver disease as a result of this disturbed hepatocyte metabolism. Reports of subacute liver failure in two SMA 1 patients receiving gene replacement treatment have further raised concerns over liver function in SMA patients. While treatment can alter the course of SMA development, there is a significantly increased risk of liver damage in those who are already at risk for metabolic disorders [111].

Chronic disease progression can result in respiratory problems, including a weak cough reflex, which increases the risk of food aspiration and inhalation when paired with reduced swallowing skills, dysphagia, and hypotonia. Therefore, the early use of artificial feeding becomes crucial in the treatment of infants with SMA 1 to provide safe breathing and sufficient nourishment. Moreover, chronic lower gastrointestinal tract constipation has been reported often and appears to have a variety of contributing factors, such as inadequate fiber and fluid intake, weak abdominal wall muscles, and gastrointestinal dysmotility. Functional dysmotility in the gastrointestinal tract, delayed colonic transit, intestinal pseudo-obstruction, or even anorectal motor dysfunction may also cause these symptoms. Some patients may feel better after taking probiotics, motility-enhancing medications, and drinking enough water. The use of high-fat meals, which are frequently used to increase calorific intake in such children, may worsen the situation by causing lower nutritional intake, raising the incidence of malnutrition [57].

Medical therapy involves both medicine and lifestyle changes (for example, a probiotic-supplemented formula based on amino acids may be suggested). Patients who do not react to conventional therapy are often not candidates for surgery. A surgical approach entails laparoscopic anti-reflux surgery and often fundoplication, which involves wrapping the stomach’s fundus around the esophagus to reestablish lower esophageal sphincter pressure. Nissen fundoplication is specifically recommended for children with a high risk of aspiration due to an absent or aberrant gag reflex, especially when pure dysphagia is present [112].

The SMN protein, which is also implicated in bone remodeling and possibly in the absorption of calcium and vitamin D, is another significant target of SMA 1 treatment. Patients with SMA frequently suffer from osteopenia, a condition in which their bones are fragile and susceptible to fractures from little trauma [111]. Prior research has demonstrated that folic acid and vitamin B12, cofactors in protein methylation, are necessary for the SMN protein function. Another advantage of protein methylation is that it can further boost binding to proteins that interact with SMNs; this binding increases in a cascade according to the degree of methylation. Protein hypomethylation can be caused by reduced intake of folic acid and vitamin B12, which could alter the severity of the SMA phenotype [111].

Children with SMA, particularly those with SMA 2, have also been found to be at serious risk for iron insufficiency. In one study, 17.5% of the entire sample population had iron insufficiency at follow-up, making it the most common micronutrient deficit [113].

## 6. Rehabilitation

To increase patient safety and autonomy, pharmacological therapy is combined with a motor rehabilitation program that uses assistive technology. Physiatrists and physical therapists should help and advise patients to help preserve muscular suppleness over time and boost levels of independence. Patients have shown greater safety and autonomy when participating in motor rehabilitation programs that use assistive devices in addition to medication.

Exercise training is essential for SMA patients because it helps maintain muscle mass, improves cardiorespiratory function, and increases physical fitness, all of which help to avoid joint contractures. In children with SMA, progressive resistance training has been demonstrated to be safe and well-tolerated [114,115,116].

Due to difficulties with secretory clearance and an increased risk of aspiration, children with SMA are more likely to experience acute respiratory decompensation.

In order to increase survival and improve the quality of life of SMA patients, these problems must be effectively managed. A key component of treating respiratory dysfunction in SMA patients is chest physical therapy. Essential methods to keep airways free and improve respiratory function include postural drainage, vibration, manual percussion, and the use of mechanical devices. A customized strategy tailored to each patient’s needs is necessary for effective chest physical therapy [117].

More complex treatments include the use of a mechanical in-exsufflator (MI-E) and intrapulmonary percussive ventilation (IPV). The MI-E simulates a cough by rapidly shifting the airway from positive to negative pressure, aiding in the removal of secretions. Although evidence for its efficacy in children with SMA is limited, MI-E remains a valuable tool in airway management. IPV delivers small bursts of air and saline mist into the lungs, creating a continuous flow that loosens trapped mucus, making it easier to expel. This technique is particularly effective in mobilizing secretions and preventing mucus plugging [117,118].

Patients with SMA frequently experience orthopedic issues, particularly scoliosis, contractures, and joint abnormalities. Scoliosis affects over half of all children with SMA, particularly those who are non-ambulatory or have lost their ability to walk. Scoliosis affects nearly all children with SMA 1 and 2, and develops in around 50% of those with SMA 3. Although their efficacy is debated, bracing and spinal fusion are common procedures used to treat scoliosis in people with SMA.

For SMA patients who can sit or walk, physiotherapy interventions like aquatic treatment are recommended. Weight relief, postural support, and improved antigravity movements are some of the most significant benefits of water for rehabilitation. These benefits make it easier to move around in the water, which is a great way to strengthen muscles and improve posture, balance, and flexibility. Typically, aquatic treatment is conducted in shallow, warm water (above 30 °C) and can involve a variety of exercises, including resistance, stability, range-of-motion, stretching, and aerobic training. This method works well for both general rehabilitation and for particular groups such as individuals with SMA [65].

To compensate for the weakness of muscles in a particular part of the body, orthoses are standardized or custom-made devices designed to support that area. They give the user a degree of autonomy. Self-initiated movement in childhood has been linked to improved feelings of competence, better environmental interaction, increased autonomy, spatial cognition, future cognitive and emotional skills, and novel coping mechanisms for environmental stressors. When standing or walking, ankle-foot orthoses, knee-ankle-foot orthoses, and hip-ankle-knee-foot orthoses are used to align the joints. For stabilizing posture and reducing undesired motion and deformity, corset braces, thoraco-lumbar-sacral orthoses, and cervical-thoraco-lumbar-sacral orthoses can also be helpful [65].

## 7. Psychological Interventions

All things considered, an interdisciplinary approach is necessary for the effective management of SMA. This strategy requires medical specialists who not only diagnose the problem but also play a critical role in educating patients and their families about treatment options and appropriate care. Since a diagnosis of SMA carries a heavy emotional burden, medical personnel must also offer families supportive therapy to help them deal with the difficulties of the disease.

SMA has a substantial psychological impact on both patients and family members at its various developmental stages. In a cross-sectional study comparing children aged 3 months to 5 years with genetically proven SMA 1, 2, or 3 and healthy controls, children with SMA were shown to exhibit higher levels of stress, anxiety, and sadness. These psychological difficulties are exacerbated by the progressive limits imposed by SMA, which impede normal childhood activities and interactions. This study emphasizes that the effects on mental health are not only immediate but also result in long-term problems that influence self-sufficiency and social development [119].

Educational experiences are also important variables that impact the psychological well-being of people with SMA. Experiencing delays in meeting academic milestones and participating in school activities is closely associated with elevated levels of anxiety and despair in SMA patients. Additionally, compared to their counterparts in standard schools, students who attended personalized schools with assistance and modifications catered to their needs showed lower levels of anxiety and sadness. These findings suggest that individualized learning environments that take into account the unique requirements of SMA patients may lessen psychological stress by creating a more welcoming and encouraging learning environment. By encouraging more engagement in educational activities despite their physical constraints, such an environment can simultaneously promote improved cognitive and social growth. Furthermore, economic concerns are particularly significant when it comes to determining the psychological effects on SMA patients and their families. There are significant financial and emotional challenges for families to overcome in caring for someone with SMA, including the need for home modifications, adapted equipment, and medical treatments. This research highlights the significant potential impact of financial assistance programs and regulations in alleviating the financial strain on families impacted by SMA, which could reduce the psychological toll on both patients and caregivers [120].

Caregivers for people with SMA, usually parents, are responsible for managing the patient’s daily life, organizing intricate treatment plans, and providing both emotional and physical assistance. Providing this care takes a significant psychological toll that affects one’s overall quality of life, has a financial impact due to job loss or diminished work capacity, and causes a great deal of emotional strain from coping with a chronic and progressive illness [121].

A lack of effective healthcare coordination can add to this stress, making caregivers more anxious and frustrated, and highlighting the critical need for better SMA care management. To help caregivers cope with the mental and physical strains of caregiving, extensive support networks are needed, such as peer support groups, psychiatric therapy, and accessible and useful resources [122]. The degree of stress that families experience is significantly increased by the severity of the condition, particularly when it comes to SMA 1, which may be accompanied by end-of-life decisions. Such circumstances can be emotionally draining, resulting in feelings of helplessness and anticipatory grief; therefore, integrated psychological support is necessary [123,124]. Siblings may also experience behavioral issues that require care away from the family environment, as they may represent an extended family burden [119,125]. Several studies have demonstrated significant promise in mindfulness training, guided meditation, and breathing techniques for improving everyday functioning and general well-being for patients and their caregivers by lowering stress, anxiety, and maladaptive thinking [126]. In order to reduce the psychological burden and avoid long-term mental health consequences for patients and their caregivers, psychological support services like counseling, peer support groups, and individual and family therapy are crucial [127].

## 8. Conclusions

By analyzing its histological background, epidemiology, pathogenesis, clinical manifestations, and diagnostic methods, this work presents the complexity of SMA.

Several other works concerning SMA, including systematic reviews, have been published recently. Unlike these recent studies [128], ours places greater emphasis on dietary and rehabilitative factors rather than highlighting the significance of newborn screening, which naturally has a major influence on the appropriate pharmaceutical intervention. Basic science approaches have been crucial for understanding the biological aspects of SMA; without them, clinical progress would have been minimal. Important progress has been made over the past few years in understanding and managing SMA, with advances in treatments such as ASOs (e.g., nusinersen), gene therapies (e.g., onasemnogene abeparvovec), and small-molecule treatments (e.g., risdiplam). However, these treatments have limitations, such as invasive administration techniques and variability of response. Currently, despite these developments, there is no available treatment that can provide a permanent cure. Another challenge is that many treatments are primarily designed for younger patients and are less effective in adults. Nutritional support, along with drug therapy, plays an important role in managing the health of affected people. A tailored rehabilitation program with a variety of care options is essential for maintaining motor function and improving quality of life. A combination of medical, psychological, and nutritional guidance, rehabilitation strategies, and supportive measures addressing the multiple needs of SMA patients will be crucial to improve the quality of life of those with this complex condition.

## Figures and Tables

**Figure 1 biology-14-00977-f001:**
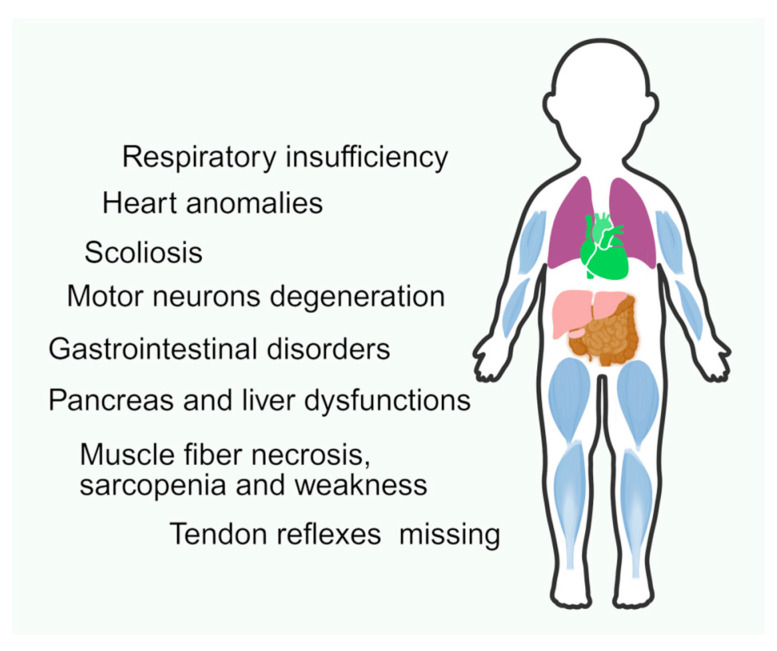
SMA pathological key aspects.

**Figure 2 biology-14-00977-f002:**
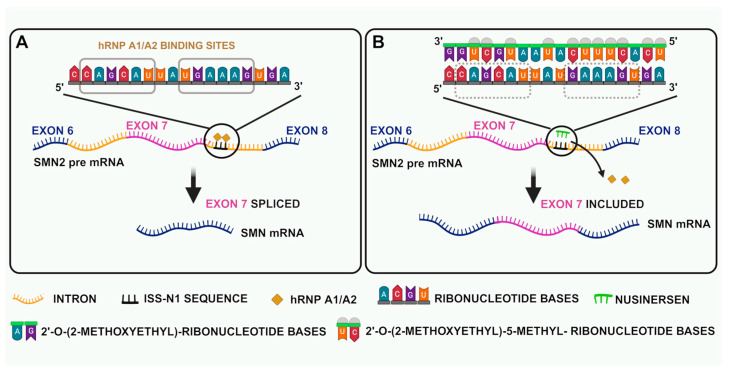
Nusinersen sequence and action. (**A**) *SMN2* not treated with Nusinersen. (**B**) *SMN2* treated with Nusinersen.

**Figure 3 biology-14-00977-f003:**
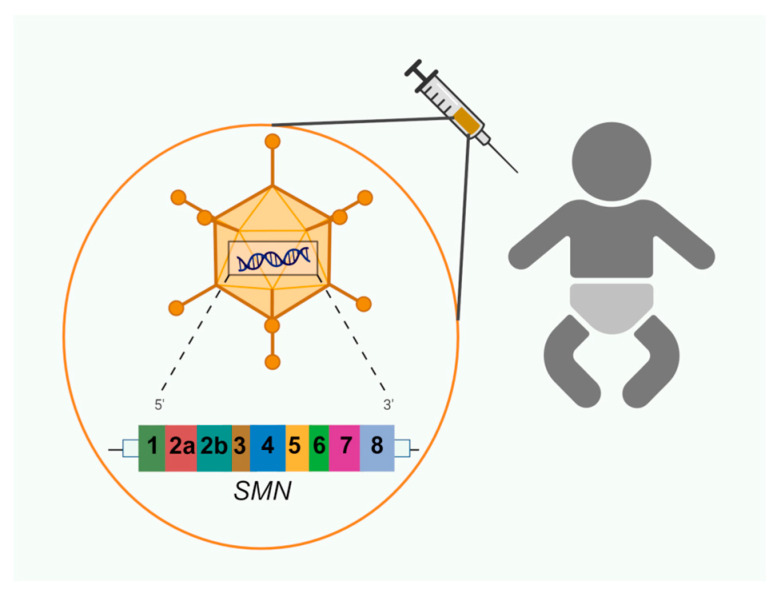
Simplified diagram of SMA gene therapy.

**Figure 4 biology-14-00977-f004:**
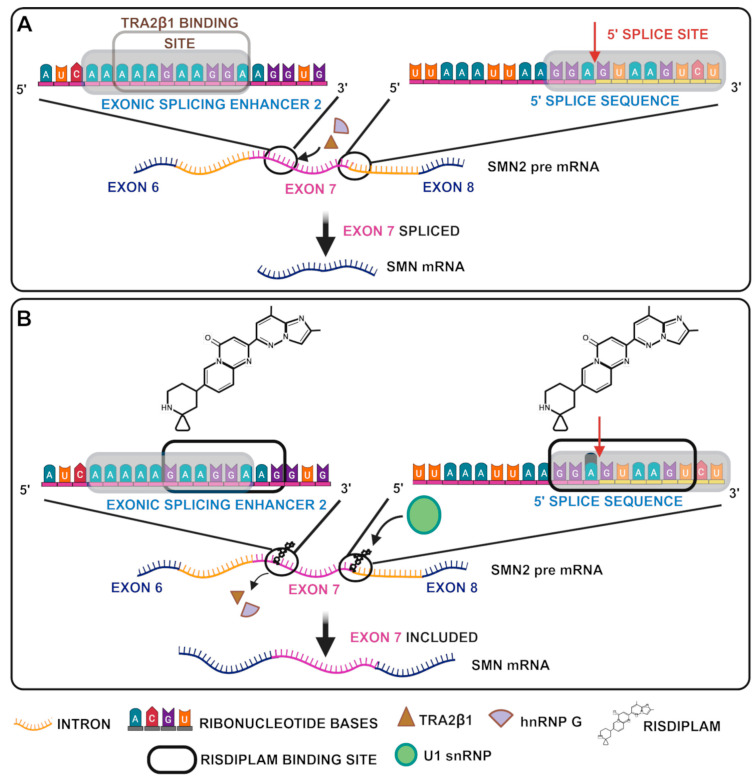
Risdiplam structure and mechanism of action. (**A**) *SMN2* not treated with Risdiplam. (**B**) *SMN2* treated with Risdiplam.

**Table 1 biology-14-00977-t001:** Schematic representation of the pathological types of SMA.

Type	Age of Onset	Max Motor Milestone	Other Motor Features	Prognosis
SMA 1A SMA 1B SMA 1C	From before birth until 2 weeks after birth 3 months 6 months	None	Severe hypotonia, 1C may reach head control	Respiratory paucity at birth, death in the first 2 weeks for 1A and by 2 years for 1B and 1C; necessary ventilation
SMA 2A SMA 2B	5 to 18 months	Sitting	Proximal weakness	Survival to adulthood
SMA 3A SMA 3B	>30 months	Walking	3A may lose the ability to walk	Normal life span
SMA 4	>30 years	Normal	Mild motor impairment	Normal life span

**Table 2 biology-14-00977-t002:** Description of the approved therapies and their current recommendations.

Treatment	Nusinersen	Onasemnogene Abeparvovec	Risdiplam
**Synthetic description**	Antisense oligonucleotide	Single-stranded *SMN1* DNA inserted into an adeno-associated virus vector	Small molecule
**Molecular action**	Modifies the splicing in the *SMN2* gene	Replace the *SMN1* non-functioning gene	Modifies the splicing in the *SMN2* gene
**Approved for**	SMA pediatric and adult	SMA 1 or up to 3 *SMN2* gene copies, up to 21 kg	SMA 1, 2, or 3, or up to 4 *SMN2* gene copies, pediatric and adult
**Administration**	Intrathecal	Slow intravenous infusion over 60 min	Enteral liquid (oral or feeding tube)
**Frequency of administration**	Four loading doses over 2 months, then every 4 months	Once	Daily

## Data Availability

No new data were created.

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
