# Peer review of "Managing Spinal Muscular Atrophy: A Look at the Biology and Treatment Strategies"

_biology, 2025, doi:10.3390/biology14080977_

Round 1
Reviewer 1 Report
Comments and Suggestions for Authors
While the paper is well-written and offers a comprehensive historical overview of various facets of Spinal Muscular Atrophy, it does not appear to introduce substantial new information to the established body of literature. To advance the field, the authors could significantly enhance their contribution by conducting a systematic review on one of the historical topics discussed, such as studies involving animal models of SMA, or the effect of disease-modifying treatments on different populations, or alternative approaches (e.g., curcumin and others). A mere compilation of presented data, however well-researched, is insufficient for a publication in Biology.
Author Response
Reviewer 1
While the paper is well-written and offers a comprehensive historical overview of various facets of Spinal Muscular Atrophy, it does not appear to introduce substantial new information to the established body of literature. To advance the field, the authors could significantly enhance their contribution by conducting a systematic review on one of the historical topics discussed, such as studies involving animal models of SMA, or the effect of disease-modifying treatments on different populations, or alternative approaches (e.g., curcumin and others). A mere compilation of presented data, however well-researched, is insufficient for a publication in Biology.
Dear Reviewer 1,
We appreciated your comment. We strongly modified the paper to accomplish your suggestions.
We hope that the revised version of the paper will fulfill the requirements for publication in Biology.
Best Regards,
Daniele Bottai
Reviewer 2 Report
Comments and Suggestions for Authors
Dr. Vezzoli et al. present a broad review of Spinal Muscular Atrophy, covering historical, genetic, phenotypic, nutritional, and therapeutic aspects. While the manuscript is timely, there are important specific issue improvements needed. Additionally, the manuscript would benefit significantly from English language and style revision. Below are the comments and questions:
The manuscript presents SMA subtypes in a table but subsequently uses alternative nomenclature, such as “Type 1.” Have the authors adopted a specific classification system that excludes this terminology? Please clarify the rationale.
What is the added value or novelty of this review compared to previous literature, particularly Schroth et al. (DOI: 10.1212/CPJ0000000000200310)? This work should be cited, along with other recent studies, and the unique contribution of the current manuscript should be clearly delineated.
The 2024 European update on SMA (Kirschner et al., Eur J Paediatr Neurol, 2024) seems relevant and should be incorporated or at least discussed.
The text refers to differences in prevalence but does not specify regions or ethnic groups. Please revise this section to include explicit information for reader clarity.
Lines 145–147
The phrase “NN are very unique cells” is too vague. Please elaborate: what specific properties or functions make motor neurons unique?
Important information is presented without citation (e.g., lines 150–152). Please ensure all factual statements are properly referenced.
Language and Style Example Line 158:
This line (and others) require revision for clarity and grammar. A full language and style edit is recommended for the entire manuscript
Lines 188–189
Which clinical or laboratory findings lead to the diagnosis? Please specify the diagnostic indicators.
Line 269 Coding of “FM”:
Did you define or code “Functional Motor” (FM)? If so, please ensure this is clearly explained.
Lines 269–271
The distinction between “functional motor” and “motor skills” is unclear. Clarify these terms and how they differ in the context of SMA, or alternatively, unify.
Line 270:
Was “FFM” also coded? Please provide definition and rationale.
Line 308–309
The phrase “Medical therapy involves both medicine…” is redundant and should be rephrased for clarity.
Line 344
The statement “The first effective medication for SMA found…” is misleading. This medication was developed, not found. Please revise.
Lines 347–349
The use of the word “modulated” is unclear. Replace it with a more precise term describing the mechanism or effect of the intervention.
Line 369
The use of “CSF” appears not clear. Does this refer to cerebrospinal fluid? Define all abbreviations at first mention.
The mechanism by which Risdiplam exerts its therapeutic effects should be more clearly and thoroughly explained.
Lines 469–470
This sentence appears disconnected and lacks coherence. Review and improve the flow, and consider rephrasing for clarity.
Line 503
The acronym “ARE” (Antioxidant Response Element) is introduced but used only once more. Codes or acronyms should be employed consistently throughout the text—ideally at least three times—to justify their inclusion.
It is important to perform a full English language and style revision.
Author Response
Reviewer 2
Dear Reviewer 2,
First of all, we would like to express our gratitude for the helpful suggestions you gave us. We are convinced that those recommendations will allow us to substantially improve our paper.
We made the changes as you indicated. Moreover, we restructure all the paper, also moving some paragraphs to different positions.
We asked Cambridge Proofreading to perform a complete revision of the paper, so we reported in the following pages the changes we made on the original paper before the English proofreading.
Dr. Vezzoli et al. present a broad review of Spinal Muscular Atrophy, covering historical, genetic, phenotypic, nutritional, and therapeutic aspects. While the manuscript is timely, there are important specific issue improvements needed. Additionally, the manuscript would benefit significantly from English language and style revision. Below are the comments and questions:
The manuscript presents SMA subtypes in a table but subsequently uses alternative nomenclature, such as “Type 1.” Have the authors adopted a specific classification system that excludes this terminology? Please clarify the rationale.
We changed the nomenclature in the text as in the table.
What is the added value or novelty of this review compared to previous literature, particularly Schroth et al. (DOI: 10.1212/CPJ0000000000200310)? This work should be cited, along with other recent studies, and the unique contribution of the current manuscript should be clearly delineated.
In order to clearly delineate the contribution of our manuscript, we introduced in the abstract:
Instead of emphasizing the importance of neonatal screening, which naturally has a significant impact on the pharmacological intervention, our approach focuses more on nutritional and rehabilitative variables.
And in the discussion:
There have been a lot of other articles recently, especially systematic reviews concerning SMA. But unlike previous studies (Schroth et al., 2024) ours places greater emphasis on dietary and rehabilitative factors rather than highlighting the significance of newborn screening, which naturally has a major influence on the pharmaceutical intervention.
The 2024 European update on SMA (Kirschner et al., Eur J Paediatr Neurol, 2024) seems relevant and should be incorporated or at least discussed.
We introduced these two sentences in the paragraph: SMA phenotypic key evidence
To identify patient populations, the abovementioned traditional SMA types by themselves are insufficient. The most significant variables that predict a patient's response to treatment in symptomatic individuals are age at onset, length of disease, and motor function level at the beginning of treatment. The most significant variables that predict a patient's response to treatment in symptomatic individuals are age at onset, length of disease, and motor function level at the beginning of treatment (Kirschner et al., 2024).
And
SMN2 copy quantity is the most significant predictor of clinical severity and age of start in patients who are genuinely presymptomatic SMA 2 patients (Kirschner et al., 2024).
The text refers to differences in prevalence but does not specify regions or ethnic groups. Please revise this section to include explicit information for reader clarity.
We revised the text:
Nonetheless, there are notable regional differences in frequency, most likely brought on by the small sample size and small patient population for this rare disease. Furthermore, a robust health system and greater knowledge of neuromuscular illnesses may be the reason for the higher incidence rate in some countries, such as Sweden. Incidence, which measures the number of new instances of the disease, varies quite a bit; on average, it should be 8 in 100,000 live birds. Furthermore, there is a great deal of variation in this parameter among nations, with some having high levels, particularly in islands like Iceland with small populations, or it may be the result of inadequate data analysis, as was the case in a study conducted in Slovakia that discovered an incidence of 17.8 in 100,000 (Verhaart et al., 2017). It may also be true, though, that various ethnic groups have varying levels of access to healthcare. A high degree of consanguinity may help to explain the high incidence (24 per 100,000 live births) seen in a small research study including 75,000 people in Libya, most likely due to a high degree of consaguinity.
Lines 145–147
The phrase “NN are very unique cells” is too vague. Please elaborate: what specific properties or functions make motor neurons unique?
We elaborated the sentence, and we introduced two references.
However, several studies show that MNs are a very unique cell type because of their size; for example, they can have very long processes that necessitate an efficient transport apparatus (Bottai and Adami, 2013; Shi et al., 2024),
Important information is presented without citation (e.g., lines 150–152). Please ensure all factual statements are properly referenced.
The reference was introduced (Zhao et al., 2021)
Language and Style Example Line 158:
This line (and others) require revision for clarity and grammar. A full language and style edit is recommended for the entire manuscript
The sentence was rewritten:
Patients with all types of SMA suffer from respiratory insufficiency, which increases along with the disease severity; deep tendon reflexes that are either missing or severely diminished; and muscular weakness. In fact, the characteristic bell-shaped chest and paradoxical breathing pattern present in the most harsh forms are caused by weakness of the intercostal muscles (Mercuri et al., 2018) .
Lines 188–189
Which clinical or laboratory findings lead to the diagnosis? Please specify the diagnostic indicators.
A new sentence was introduced:
SMN2 copy quantity is the most significant predictor of clinical severity and age of start in patients who are genuinely presymptomatic SMA 2 patients (Kirschner et al., 2024).
Line 269 Coding of “FM”:
Did you define or code “Functional Motor” (FM)? If so, please ensure this is clearly explained.
The acronym was substituted with the full word fat mass.
Lines 269–271
The distinction between “functional motor” and “motor skills” is unclear. Clarify these terms and how they differ in the context of SMA, or alternatively, unify.
With the previous correction, we think that now the sentence is clearer.
Line 270:
Was “FFM” also coded? Please provide definition and rationale.
The acronym was substituted with the full word fat-free mass.
Line 308–309
The phrase “Medical therapy involves both medicine…” is redundant and should be rephrased for clarity.
The redundancy was removed.
Line 344
The statement “The first effective medication for SMA found…” is misleading. This medication was developed, not found. Please revise.
The revision was made.
Lines 347–349
The use of the word “modulated” is unclear. Replace it with a more precise term describing the mechanism or effect of the intervention.
The sentence was changed as reported:
Nusinersen is an ASO that induces exon 7 inclusion by binding to the intronic splicing inhibitor in intron 7 in the mRNA transcripts of survival SMN2 (Corey, 2017).
Line 369
The use of “CSF” appears not clear. Does this refer to cerebrospinal fluid? Define all abbreviations at first mention.
CSF was substituted with the full words.
The mechanism by which Risdiplam exerts its therapeutic effects should be more clearly and thoroughly explained
A new descriptive sentence and reference were introduced
Lines 469–470
This sentence appears disconnected and lacks coherence. Review and improve the flow, and consider rephrasing for clarity.
We remove the refused sentence
Line 503
The acronym “ARE” (Antioxidant Response Element) is introduced but used only once more. Codes or acronyms should be employed consistently throughout the text—ideally at least three times—to justify their inclusion.
ARE acronym was removed.
We hope that the revised version of the paper will fulfill the requirements for publication in Biology.
Best Regards,
Daniele Bottai
Reviewer 3 Report
Comments and Suggestions for Authors
Comments on Vezzoli et al.
In their review entitled "Managing spinal muscular atrophy: A look at the biology and treatment strategies" the authors provide an overview over the biological background of SMA, clinical phenotypes, treatment demands and treatment options both established and under development. The review adds only very little relative to excellent overviews recently published in this field. Its strength lies in the clinical perspective on the disease, but it falls short of discussing this perspective in relation to the novel situation generated by the availability of therapies that transformed SMA from an often deadly to a manageable neurodegenerative disease.
The following points should be addressed by the authors:
- The whole abstract needs to be completely restructured/rewritten to convey the main points of the review in a clear and conclusive way.
- The chapter addressing key genetic evidence and the basic science parts of the following chapter need to be generally improved. In addition, the authors should also address a number of erroneous statements as detailed below.
- Lines 44-46: the statement that most of the functions of SMN concern mRNA splicing is an oversimplification. Indeed, many other functions of the SMN have been unraveled and it is not clear which ones are critical for the pathogenesis of SMA!
- Lines 69-70: please define the abbreviations FM and FFM!
- Line 85: SMN is not a component of mature spliceosomes, but a cofactor needed for the biogenesis of some snRNPs as detailed in the following sentences of the manuscript!
- Line 87: the SMN protein is contained in a complex called "SMN protein complex" and not "gemins". This complex does not only consist of gemins 2-8 but also contains the protein UNRIP. Furthermore, SMN/the SMN complex has been demonstrated to interact with many more proteins (and RNA) on a temporary basis, e.g. with hnRNPQ/R, Z-DNA binding protein or proteins of the cytoskeleton like actin and profilin.
- Line 94: mice do not have only one copy of the SMN gene per diploid chromosome but two! Please correct!
- Line 95: the statement "SMN gene also expresses the same SMN protein" is misleading because only about 10-15 % of the mature SMN2 mRNA contain exon 7 and are translated into full length protein. 85-90 % of the mature SMN2 mRNA does not contain exon 7 and thus gives rise to truncated SMN2 protein that is rapidly degraded.
- Line 153: the proteins a-actinin and a-B crystallin are not unique to skeletal muscle as stated: a-actinin occurs in the spine apparatus of hippocampal neurons for instance and a-B crystallin is present in the eyes as well.
- Page 6 onward: the authors discuss the traditional phenotypic SMA classification scheme. Although still helpful, with the access to causal therapy in most developed countries this has become somewhat obsolete. Another scheme based on newborn screening and therapy options has recently proposed (Varone et al. (2025) Spinal muscular atrophy in the era of newborn screening: how the classification could change, Front. Neurol. 16:1542396. doi: 10.3389/fneur.2025.1542396.
- Fig. 2: What do the beige half-circles on the RNA denote? Are these the chemical modifications mentioned above in line 348? Please label properly!
- Figs. 2 and 4 are somewhat under-complex. The authors explain in the text that Nusinersen functions as a competitor for protein factors interacting with ISS-N1. Why not depicting exactly that? The same applies to Fig. 4: the target of Risdiplam is known, why not integrating it into the graphics to show the "mechanism" as stated in the legend?
- Line 363: The primary sites of damage are not only a-motorneurons in the spinal cord but also in the brain stem! The term "main site of damage" that is used by the authors is misleading, because secondary sites of damage like the degenerating skeletal muscles and other tissues possibly affected by low SMN levels may account for much greater damage!
- Lines 477-480 and lines 536 to 540 contain redundant information on RO7204239. These two blocks should be combined, perhaps with only a short note at the second site referring to the previous one.
- Line 499: it remains unclear why the absence of exon 7 from the SMN protein should render this a "sensor" for oxidative stress. The authors should detail a possible mechanism and add available citations!
- Lines 551+552: Citation is missing!
- Line 572: Stasimon overexpression did not revert the SMA phenotype (Wirth, B. et al. (2020) Rev. Genom. Hum. Genet. 21: 231– 61 and citations therein)
Minor points/Typos:
- Line 473: fellows
- Line 476: zolgensma
Author Response
Dear Reviewer 3,
First of all, we would like to express our gratitude for the helpful suggestions you gave us. We are convinced that those recommendations will allow us to substantially improve our paper.
We made the changes as you indicated. Moreover, we restructure all the paper, also moving some paragraphs to different positions.
We asked Cambridge Proofreading to perform a complete revision of the paper, so we reported in the following pages the changes we made on the original paper before the English proofreading.
Comments on Vezzoli et al.
In their review entitled "Managing spinal muscular atrophy: A look at the biology and treatment strategies" the authors provide an overview over the biological background of SMA, clinical phenotypes, treatment demands and treatment options both established and under development. The review adds only very little relative to excellent overviews recently published in this field. Its strength lies in the clinical perspective on the disease, but it falls short of discussing this perspective in relation to the novel situation generated by the availability of therapies that transformed SMA from an often deadly to a manageable neurodegenerative disease.
The following points should be addressed by the authors:
- The whole abstract needs to be completely restructured/rewritten to convey the main points of the review in a clear and conclusive way.
We retructured the abstract.
- The chapter addressing key genetic evidence and the basic science parts of the following chapter need to be generally improved. In addition, the authors should also address a number of erroneous statements as detailed below.
- Lines 44-46: the statement that most of the functions of SMN concern mRNA splicing is an oversimplification. Indeed, many other functions of the SMN have been unraveled and it is not clear which ones are critical for the pathogenesis of SMA!
This sentence was introduced with relative reference:
In the animal kingdom, SMN protein plays a multifunctional role that is essential for to survival. SMN affects a number of facets of RNA metabolism controls the synthesis of signal recognition particles, telomerase, small nuclear RNPs, small nucleolar RNPs, and small Cajal body-associated RNPs. Additionally, transcription, pre-mRNA splicing, histone mRNA processing, translation, selenoprotein synthesis, macromolecular trafficking, stress granule formation, cell signaling, cytoskeleton maintenance, and DNA repair are all significantly impacted by SMN (Singh et al., 2017)
- Lines 69-70: please define the abbreviations FM and FFM!
The acronyms were substituted with their full words, "fat mass" and "fat-free mass."
- Line 85: SMN is not a component of mature spliceosomes, but a cofactor needed for the biogenesis of some snRNPs as detailed in the following sentences of the manuscript!
This sentence was deleted.
- Line 87: the SMN protein is contained in a complex called "SMN protein complex" and not "gemins". This complex does not only consist of gemins 2-8 but also contains the protein UNRIP. Furthermore, SMN/the SMN complex has been demonstrated to interact with many more proteins (and RNA) on a temporary basis, e.g. with hnRNPQ/R, Z-DNA binding protein or proteins of the cytoskeleton like actin and profilin.
This sentence was modified, and new references were introduced.
The SMN protein is part of a bigger complex called Gemins, which is made up of a number of other proteins (Gemin2–8). UNRIP (Fallini et al., 2012). Additionally, it has been shown that SMN and the SMN complex may interact momentarily with a wide range of other proteins (including RNA), such as hnRNPQ/R (Rossoll et al., 2002), and cytoskeleton proteins like actin and profilin (Bowerman et al., 2007; Singh et al., 2017).
Line 94: mice do not have only one copy of the SMN gene per diploid chromosome but two! Please correct!
The sentence was corrected.
- Line 95: the statement "SMN gene also expresses the same SMN protein" is misleading because only about 10-15 % of the mature SMN2 mRNA contain exon 7 and are translated into full length protein. 85-90 % of the mature SMN2 mRNA does not contain exon 7 and thus gives rise to truncated SMN2 protein that is rapidly degraded.
The sentence was rewritten:
Chimpanzees contain two to seven copies of the SMN gene per diploid genome, according to studies (Germain-Desprez et al., 2001), while other species, such as mice, only have two copies (Wirth et al., 2020). In humans, there are just eleven nucleotide differences between the SMN1 and SMN2 genes. Remarkably, a C6T inversion in exon 7 functions as an exonic enhancer but has no effect on the amino acid sequence causing the production of a shorter form lacking of exon 7 (Farrar and Kiernan, 2015) in around 90% of instances. About 10% of the SMN2 gene is translated into a full-length protein that can partially compensate for SMN1's lack (Adami and Bottai, 2019).
- Line 153: the proteins a-actinin and a-B crystallin are not unique to skeletal muscle as stated: a-actinin occurs in the spine apparatus of hippocampal neurons for instance and a-B crystallin is present in the eyes as well.
The words “unique to muscle” were removed.
- Page 6 onward: the authors discuss the traditional phenotypic SMA classification scheme. Although still helpful, with the access to causal therapy in most developed countries this has become somewhat obsolete. Another scheme based on newborn screening and therapy options has recently proposed (Varone et al. (2025) Spinal muscular atrophy in the era of newborn screening: how the classification could change, Front. Neurol. 16:1542396. doi: 10.3389/fneur.2025.1542396.
The suggested classification was introduced.
Given that the majority of developed countries already have access to causative therapy, this definition is a little outdated. A new classification that focuses on newborn screening and treatment options was recently released (Varone et al., 2025). This new approach takes advances from the newbor screening. These authors divided the patients into three classes:
- Clinically undetermined and genetically determined;
- Clinically and genetically determined infants;
- Clinically determined but genetically undetermined infants;
and suggested the appropriate pharmacological treatment between nusinersen, onasemnogene abeparvovec, and risdiplam.
- 2: What do the beige half-circles on the RNA denote? Are these the chemical modifications mentioned above in line 348? Please label properly!
The label was properly made.
- 2 and 4 are somewhat under-complex. The authors explain in the text that Nusinersen functions as a competitor for protein factors interacting with ISS-N1. Why not depicting exactly that? The same applies to Fig. 4: the target of Risdiplam is known, why not integrating it into the graphics to show the "mechanism" as stated in the legend?
The two figures were redrawn as suggested by the reviewer.
- Line 363: The primary sites of damage are not only a-motorneurons in the spinal cord but also in the brain stem! The term "main site of damage" that is used by the authors is misleading, because secondary sites of damage like the degenerating skeletal muscles and other tissues possibly affected by low SMN levels may account for much greater damage!
The sentence was corrected.
The MNs in the spinal cord and the brain stem are the first cells hit in SMA.
- Lines 477-480 and lines 536 to 540 contain redundant information on RO7204239. These two blocks should be combined, perhaps with only a short note at the second site referring to the previous one.
These two sentences were combined:
RO7204239 is a monoclonal anti-myostatin that was developed and studied by Hoffmann-La Roche study is in phase 3 trial. It is a very potent antibody that targets human latent myostatin and inhibits its conversion to the active form when administered subcutaneously. This muscle- directed treatment was administered in combination with risdiplam (RO7034067) to investigate the safety and efficacy in SMA patients with walking capabilities (MANATEE study, NCT05115110).
- Line 499: it remains unclear why the absence of exon 7 from the SMN protein should render this a "sensor" for oxidative stress. The authors should detail a possible mechanism and add available citations!
The sentence was changed for clarity:
Inducing oxidative stress in an animal model of SMA, transgenic (TG) mice with FVB/N background, which have two copies of SMN2 and one copy of Smn (Smn+/-;SMN2+/+), Seo and colleagues (Seo et al., 2016) suggested that several cis-elements and transacting factors are implicated in the splicing control of distinct SMN exons under normal and OS settings, as well as the susceptibility of different SMN2 exons to skipping (including the 7) under OS conditions
- Lines 551+552: Citation is missing!
The reference was introduced (Kim et al., 2017).
- Line 572: Stasimon overexpression did not revert the SMA phenotype (Wirth, B. et al. (2020) Rev. Genom. Hum. Genet. 21: 231– 61 and citations therein)
The sentence was corrected and the reference introduced.
The following reduction of Stasimon, a transmembrane protein found in the endoplasmic reticulum, due to SMN-mediated U12 (an snRNA gene) intron splicing may be linked to an activation of p53.However, Stasimon overexpression did not revert the SMA phenotype (Wirth et al., 2020)
Minor points/Typos:
- Line 473: fellows
The word was corrected.
- Line 476: zolgensma
The word was corrected.
We hope that the revised version of the paper will fulfill the requirements for publication in Biology.
Best Regards,
Daniele Bottai
Reviewer 4 Report
Comments and Suggestions for Authors
The authors summarize the biological background of and therapeutic options in SMA
First, English should be much improved, specifically in the first half of the manuscript. E.g. in Simple summary: „SMA is a dangerous disease” should be replaced (e.g. severe or serious). In line 17, “SMA was noticed…” should also replace, in line 74 delete “sometimes stated”
Line 129-130 can be deleted, because assessment of the SMN2 gene is important not simple for the diagnosis, but instead, for prognosis and treatment. It is described elsewhere in the text.
Line 159: how are deep tendon reflexes related to chest deformity and respiratory insufficiency ?
Lines 269, 290: what is FM and FFM?
Lines 363-364: please explain that intrathecal delivery is necessary because nusinersen does not cross blood-brain barrier due to its high molecular weight
Line 370. Here you can indicate, that the usual treatment interval is 120 days because of the elimination time
line 419. After you mentioned the necessary amount of vector, you could mention the age and weight limit of this therapy
line 452: in risdiplam chapter you should mention its limitations, including again the weight
Line 524: delete “alpha”
Generally, at the end of chapter 4, a table containing the approved and the under investigation therapies, might help the reader to overview the therapeutic options.
Furthermore, here authors should summarize the current recommendations for SMA therapy.
Comments on the Quality of English LanguageEnglish quality should be improved.
Author Response
Dear Reviewer 4
First of all, we would like to express our gratitude for the helpful suggestions you gave us. We are convinced that those recommendations will allow us to substantially improve our paper.
We made the changes as you indicated. Moreover, we restructure all the paper, also moving some paragraphs to different positions.
We asked Cambridge Proofreading to perform a complete revision of the paper, so we reported in the following pages the changes we made on the original paper before the English proofreading.
The authors summarize the biological background of and therapeutic options in SMA
First, English should be much improved, specifically in the first half of the manuscript. E.g. in Simple summary: „SMA is a dangerous disease” should be replaced (e.g. severe or serious). In line 17, “SMA was noticed…” should also replace, in line 74 delete “sometimes stated”
We changed all the points.
Line 129-130 can be deleted, because assessment of the SMN2 gene is important not simple for the diagnosis, but instead, for prognosis and treatment. It is described elsewhere in the text.
The sentence was deleted.
Line 159: how are deep tendon reflexes related to chest deformity and respiratory insufficiency ?
The sentence was changed.
Patients with all types of SMA suffer from respiratory insufficiency, which increases along with the disease severity; deep tendon reflexes that are either missing or severely diminished; and muscular weakness. In fact, the characteristic bell-shaped chest and paradoxical breathing pattern present in the most harsh forms are caused by weakness of the intercostal muscles (Mercuri et al., 2018) .
Lines 269, 290: what is FM and FFM?
The acronyms were substituted with ther full words: fat mass and fat-free mass.
Lines 363-364: please explain that intrathecal delivery is necessary because nusinersen does not cross blood-brain barrier due to its high molecular weight
We inserted this sentence:
since it cannot pass the blood-brain barrier due to its high molecular weight.
Line 370. Here you can indicate, that the usual treatment interval is 120 days because of the elimination time
We changed the sentence as:
Nusinersen is metabolized via hydrolysis catalyzed by exonucleases. In the cerebrospinal fluid, the typical terminal elimination half-life is predicted to be between 135 and 177 days; for this reason, the administration interval is 120 days.
line 419. After you mentioned the necessary amount of vector, you could mention the age and weight limit of this therapy
The sentence was changed as:
Unquestionably, clinical trials have demonstrated the effectiveness of onasemnogene abeparvovec in treating babies with SMA who are between the ages of two weeks and eight months and weigh less than 13.5 kg (Ogbonmide et al., 2023); however, recent analysis demonstrated that in a UK population, zolgensma was found to be effective in a population between 3.2 and 20.2 kg and aged between 0.6 and 89 months (Gowda et al., 2024).
line 452: in risdiplam chapter you should mention its limitations, including again the weight
This sentence was introduced:
On the basis of Roche guidelines, risdiplam ca be assumen from age lower than to months (0.15 mg/kg/day), in a liquid formulation, up to more than 2 year (more than 20 kg of weigh and the dosage of 5 mg/day in liquid o tablet formulation) this dosage can be administered to adult patients (https://www.evrysdi-hcp.com/content/dam/gene/evrysdi-hcp/pdf/evrysdi-hcp-dosing-guide.pdf).
Line 524: delete “alpha”
Alpha was delited
Generally, at the end of chapter 4, a table containing the approved and the under investigation therapies, might help the reader to overview the therapeutic options.
Furthermore, here authors should summarize the current recommendations for SMA therapy.
A table including a summary and a recommendation for SMA therapy was inserted
We hope that the revised version of the paper will fulfill the requirements for publication in Biology.
Best Regards,
Daniele Bottai
Round 2
Reviewer 1 Report
Comments and Suggestions for Authors
While this new version includes minor textual improvements, it unfortunately hasn't altered my overall assessment from version 1. The paper is commendably well-written and provides a comprehensive historical overview of various aspects of Spinal Muscular Atrophy. However, its primary limitation remains the lack of substantial new information or a novel contribution to the existing body of literature. To elevate this work and make a significant impact in the field, I strongly recommend the authors enhance their contribution by conducting a systematic review on one of the historical topics already discussed within the paper. This would transform the work from a compilation into a more rigorous and publishable piece of research.
Author Response
Dear Reviewer 1,
We understand your point; for that reason, in version R1, we restructure the paper in order to better focus our work following our original intention when we started to write the manuscript.
There have been several recent publications on SMA, including many systematic reviews. Instead of emphasizing the importance of newborn screening, which naturally has a significant impact on the appropriate pharmaceutical intervention, our study focuses more on dietary and rehabilitative factors than these recent studies.
One of the main focuses of our work was the dietary approach. Indeed, we recently published a work on the effect of curcumin on neural stem cells in a mouse model of SMA, and we proposed this approach as an example since the outcome of the paper was really intriguing.
Furthermore, we focus on rehabilitation. Indeed, preserving motor function and enhancing quality of life require a customized rehabilitation program with a range of care alternatives. To enhance the quality of life for people with SMA, a mix of medical, psychological, and nutritional advice, rehabilitation techniques, and supporting measures that suit the many requirements of SMA patients will be essential.
We apologize if we were unable to fully satisfy your wishes, and we really hope that you will accept the work after reading this clarification and the updated revisions we made.
Warm regards,
Daniele Bottai
Reviewer 2 Report
Comments and Suggestions for Authors
After remodeling, the manuscript is now acceptable for publication
Author Response
We thank Reviewer 2 for his/her help in improving our work.
Best regards,
Daniele Bottai
Reviewer 3 Report
Comments and Suggestions for Authors
Version 2 of the manuscript is clearly improved and also profits from the external language editing. In some instances, however, the editing mutilated sentences and destroyed the meaning. The authors appear to have missed these negative changes. Apart from that, there are still some minor complains and an error in altered Fig. 4 that should be addressed.
In the abstract, the first two sentences focus on the impact of SMA, but still are a bit redundant. They should be combined.
Abstract, line28-31: what do the authors mean with the sentence "...for those who do not respond to newly approved drugs and older neuron systems"? This obviously is an editing error because in version 1 this read: (for) "old" patients that already have compromised most of the motor neurons".
Line 131: to this reviewer's knowledge the first experimental evidence for a direct interaction between SMN and actin has been provided only very recently! For reference, see Schuning et al. (2024) FASEB J. 38, e70055, doi: 10.1096/fj.202300183R.
Line 139: a C to T mutation is called "transition" and not "inversion". This should be corrected!
Line 144: "Subtle alterations on the second allele of chromosome 5" probably should read: "Subtle alterations of the second allele on chromosome 5?
Line 155: "Since early care...": this is somewhat imprecise, the authors probably mean: "early start of treatment"?
line 211 onwards: please label SMA types consistently in one way! In version 2 of the manuscript at least 3 distinct forms are used SMA-1, SMA 1 and SMA-1!
In Fig. 4 A and B the direction of mRNA has been erroneously labeled 3' to 5' (Exon 6 to 8). Please correct! Furthermore, the term 5' splice site does not refer to exon 7 but to the intron following exon 7!
Lines 461 to 468: this piece of information on new trials on nusinersen and risdiplam is interspersed between two blocks about anti-myostatin approaches. It would appear to be a more logical order if this block of information is placed before the two anti-myostatin blocks.
Lines 528-530: this sentence is redundant to the paragraph above (lines 521-527) and should be combined
Lines 539-541: this sentence has lost its verb in the editing process!
Line 595: please correct: growth hormone releasing hormone receptor (growth is lacking).
Author Response
Dear Reviewer 3,
First of all, we would like to thank you for your patience and for your efforts in helping us in improving your work.
We answer you point by point:
Version 2 of the manuscript is clearly improved and also profits from the external language editing. In some instances, however, the editing mutilated sentences and destroyed the meaning. The authors appear to have missed these negative changes.
We went through the paper, and we corrected these negative editing changes:
Specifically:
Line 144: "Subtle alterations on the second allele of chromosome 5" probably should read: "Subtle alterations of the second allele on chromosome 5?
We thank Reviewer 3 for the correction, this was one of the external language editing errors that we did not correct.
Line 224
We thank Reviewer 3 for the hint, this was one of the external language editing errors that we did not correct, the part in yellow was reintroduced
The most common variety of spinal muscular atrophy is SMA 1 (Werdnig–Hoffmann disease), which accounts for around 50% of all instances of the ailment that are identified (Arnold et al., 2015).
Line 360
We thank Reviewer 3 for the hint, this was one of the external language editing errors that we did not correct, the part in yellow was reintroduced
Although gGene therapy has many inherent obstacles, one of the most significant is the challenge of getting the gene into the cell.
Line 499
We thank Reviewer 3 for the hint, this was one of the external language editing errors that we did not correct, the part in yellow was reintroduced
Seo and colleagues induced oxidative stress in an animal model of SMA, using namenly a transgenic (TG) mice with FVB/N background, which have two copies of SMN2 and one copy of Smn (Smn+/-;SMN2+/+) (Seo et al., 2016).
Lines 539-541: this sentence has lost its verb in the editing process!
Also was one of the external language editing errors that we did not correct.
In mild SMA models, notable gains in muscle mass and functional ability were demonstrated after recent developments in gene therapy employing AAV vectors to carry soluble ActRIIB or myostatin pro-peptides resistant to proteolytic activation
Line 601
We thank Reviewer 3 for the hint, this was one of the external language editing errors that we did not correct, the part in yellow was reintroduced
The ubiquitin-activating enzyme E1, encoded by UBA1, is an essential part of the ubiquitin-proteasome system (UBA), which breaks down undesirable or damaged cellular proteins (Chaytow et al., 2021).
Apart from that, there are still some minor complains and an error in altered Fig. 4 that should be addressed.
In the abstract, the first two sentences focus on the impact of SMA, but still are a bit redundant. They should be combined.
The first two sentences were combined:
Since its discovery in the late 19th century, spinal muscular atrophy (SMA) had a significant medical and societal impact, primarily affecting newborns, toddlers, and young adults.
Abstract, line28-31: what do the authors mean with the sentence "...for those who do not respond to newly approved drugs and older neuron systems"? This obviously is an editing error because in version 1 this read: (for) "old" patients that already have compromised most of the motor neurons".
We thank the Reviewer 3 for noticing this refusal, we change back the sentence
Treatments are needed for those who do not respond to newly approved drugs and older patients with significantly compromised neuron systems.
Line 131: to this reviewer's knowledge the first experimental evidence for a direct interaction between SMN and actin has been provided only very recently! For reference, see Schuning et al. (2024) FASEB J. 38, e70055, doi: 10.1096/fj.202300183R.
We thank Reviewer 3 for the focused indication; we added the reference Schuning et al. (2024) FASEB J. 38, e70055, doi: 10.1096/fj.202300183R
Line 139: a C to T mutation is called "transition" and not "inversion". This should be corrected!
The word was corrected
Line 144: "Subtle alterations on the second allele of chromosome 5" probably should read: "Subtle alterations of the second allele on chromosome 5?
We thank Reviewer 3 for the correction; this was one of the external language editing errors that we did not correct.
Line 155: "Since early care...": this is somewhat imprecise, the authors probably mean: "early start of treatment"?
The sentence was changed as suggested
Since early start of treatment greatly improves…
line 211 onwards: please label SMA types consistently in one way! In version 2 of the manuscript at least 3 distinct forms are used SMA-1, SMA 1 and SMA-1!
We went through the paper searching for SMA, and we unified the nomenclature. We could not find SMA-1.
In Fig. 4 A and B the direction of mRNA has been erroneously labeled 3' to 5' (Exon 6 to 8). Please correct! Furthermore, the term 5' splice site does not refer to exon 7 but to the intron following exon 7!
The figure was modified according to reviewer 3's suggestions: We underline that the intron that follows exon 7 is indicated in yellow, and indeed the sequence indicated as 5’ splice includes part of exon 7 and part of the next intron. In order to make the picture more intuitive, we did a slight modification of it.
Lines 461 to 468: this piece of information on new trials on nusinersen and risdiplam is interspersed between two blocks about anti-myostatin approaches. It would appear to be a more logical order if this block of information is placed before the two anti-myostatin blocks.
This chapter was reorganized as reviewer 3 suggested.
Lines 528-530: this sentence is redundant to the paragraph above (lines 521-527) and should be combined
The two sentences were combined:
A number of other studies conducted in recent years have focused on muscle loss treatments aimed at blocking myostatin (growth differentiation factor 8; GDF-8), a protein that negatively regulates muscle growth. As a member of the transforming growth factor-β superfamily, myostatin may contribute to muscle atrophy and wasting and could be an accessible therapeutic target for conditions like SMA. Biohaven Pharmaceuticals is now researching the effectiveness of taldefgrobep-α, a new myostatin inhibitor, in enhancing muscle mass and strength in SMA patients.
Lines 539-541: this sentence has lost its verb in the editing process!
Also was one of the external language editing errors that we did not correct.
In mild SMA models, notable gains in muscle mass and functional ability was demonstrated after recent developments in gene therapy employing AAV vectors to carry soluble ActRIIB or myostatin pro-peptides resistant to proteolytic activation
Line 595: please correct: growth hormone releasing hormone receptor (growth is lacking).
The term growth was inserted
We hope that this updated version of the work is good enough to be published in Biology.
Sincerely,
Bottai Daniele

Reviewer 4 Report
Comments and Suggestions for Authors
Authors addressed all review points and significantly improved the manuscript.
Author Response
We thank Reviewer 4 for his/her help in improving our work.
Best regards,
Daniele Bottai